# Investigating the Molecular Mechanisms Underlying Early Response to Inflammation and *Helicobacter pylori* Infection in Human Gastric Epithelial Cells

**DOI:** 10.3390/ijms242015147

**Published:** 2023-10-13

**Authors:** Giulia Martinelli, Marco Fumagalli, Stefano Piazza, Nicole Maranta, Francesca Genova, Paola Sperandeo, Enrico Sangiovanni, Alessandra Polissi, Mario Dell’Agli, Emma De Fabiani

**Affiliations:** Department of Pharmacological and Biomolecular Sciences “Rodolfo Paoletti”, University of Milan, 20133 Milan, Italy; giulia.martinelli@unimi.it (G.M.); marco.fumagalli3@unimi.it (M.F.); nicole.maranta@unimi.it (N.M.); genova.francesca@hsr.it (F.G.); paola.sperandeo@unimi.it (P.S.); enrico.sangiovanni@unimi.it (E.S.); alessandra.polissi@unimi.it (A.P.); mario.dellagli@unimi.it (M.D.); emma.defabiani@unimi.it (E.D.F.)

**Keywords:** *Helicobacter pylori*, GES-1 cells, AGS cells, IL-8, IL-6, NF-κB, CEBPβ

## Abstract

*Helicobacter pylori* is a leading cause of chronic gastric inflammation, generally associated with gastritis and adenocarcinoma. Activation of the NF-κB pathway mainly contributes to the inflammatory phenotype observed in *H. pylori* infection in humans and experimental models. Since the gastric epithelium undergoes rapid turnover, inflammation and pathogenicity of *H. pylori* result from early phase and chronically activated pathways. In the present study we investigated the early host response to *H. pylori* in non-tumoral human gastric epithelial cells (GES-1). To dissect the pathogen-specific mechanisms we also examined the response to tumor necrosis factor (TNF), a prototypical cytokine. By analyzing the activation state of NF-κB signaling, cytokine expression and secretion, and the transcriptome, we found that the inflammatory response of GES-1 cells to *H. pylori* and TNF results from activation of multiple pathways and transcription factors, e.g., NF-κB and CCAAT/enhancer-binding proteins (CEBPs). By comparing the transcriptomic profiles, we found that *H. pylori* infection induces a less potent inflammatory response than TNF but affects gene transcription to a greater extent by specifically inducing transcription factors such as CEBPβ and numerous zinc finger proteins. Our study provides insights on the cellular pathways modulated by *H. pylori* in non-tumoral human gastric cells unveiling new potential targets.

## 1. Introduction

It is estimated that *Helicobacter pylori* successfully colonizes the human gastric epithelium of about 50% of the entire world population [1]. Moreover, the association between *H. pylori* infection and development of gastric carcinoma has been well documented, leading the World Health Organization to classify *Helicobacter pylori* as a human carcinogen. In line with this, in 2021 chronic infection with *H. pylori* was confirmed in the list of substances that are known or reasonably anticipated to cause cancer in humans in the National Toxicology Program’s 15th Report on Carcinogens [2].

It has long been known that *H. pylori* infection leads to NF-κB activation and, consequently, to expression of IL-8 either in model systems or in clinical settings [3,4]. Whether the activation of the NF-κB pathway upon *H. pylori* infection strictly depends on CagA has been debated. It has been shown that *H. pylori* activates NF-κB by engaging a specific signaling pathway that comprises bacterial-derived effector metabolites, i.e., D-glycero-b-D-manno-heptose 1,7-bisphosphate, and host proteins such as ALPK1 kinase and TIFA proteins [5]. Of note, this machinery is not involved in the activation of NF-κB induced by proinflammatory cytokines such as tumor necrosis factor (TNF) or interleukin 1β (IL-1β).

The type IV secretion system (T4SS) is crucial for delivery of virulence factors from *H. pylori* to host cells. It is encoded by the Cag pathogenicity island and is strongly associated with increased risk of developing gastric adenocarcinoma in infected subjects [6]. The pathogenetic role of T4SS in gastric cancer is linked to the fact that it mediates the translocation of the virulence factor CagA, the only effector protein known to be secreted by the *H. pylori* Cag T4SS, into host cells [7]. Structural component of the Cag T4SS, such as CagL, have been implicated not only in CagA release, but also in direct inflammatory signals [8]. In addition, *H. pylori* T4SS is also involved in the release of the pathogen-derived metabolite D-glycero-beta-D-manno-heptose 1,7-bisphosphate mentioned above [5]. As for CagA, it is responsible for alteration of various signaling cascades and aberrant activation of oncogenic proteins such as Ras, β-catenin, phosphatidylinositol-3-kinase, and others [9].

Most of the evidence regarding the pathogenicity of *H. pylori* and the underlying molecular mechanisms has been obtained in gastric cancer cell lines, e.g., AGS, MKN45, and KATO III, to name a few. Although these cell lines present several advantages, they are not fully representative of the conditions occurring in *H. pylori*-infected stomach mucosa. Gastric organoids probably represent the in vitro model that most closely resembles whole organ physiology [10]. Nevertheless, conventional cell lines can still be considered a convenient system for applications such as screening of compounds with potential therapeutic activity and proof of concept validation of pathogenetic mechanisms. In this regard, GES-1 cells have been used by several groups, including ours, to test potential anti-gastritis agents (see for example references [11,12,13,14]). Furthermore, the effects of *H. pylori* on genomic instability and expression of oncogenes have been investigated in GES-1 cells [15,16]; nevertheless, an in-depth characterization of the pathways triggered in this cell line in response to inflammatory stimuli and pathogens is still incomplete.

It has been recently estimated that normal stomach epithelial cells have a turnover of about 10 days [17]. Moreover, *H. pylori* infection increases cell proliferation rate of normal epithelial cells as demonstrated in gastric organoid cultures [18,19]. Thus, it is conceivable that also in conditions of chronic *H. pylori* infection, a significant fraction of epithelial cells is continuously replaced with new cells that become exposed to the pathogen for the first time. In this context, factors and pathways triggered in the early phases of infection probably overlap with those occurring in the chronic phase, thus contributing altogether to the activation of pathogenetic pathways. From the therapeutic point of view, treatments targeting early-phase in addition to chronically activated pathways would be more effective.

The aim of the present study was to characterize the early-stage host response in normal gastric GES-1 cells infected with *H. pylori*, in comparison to the tumoral AGS cell line. As non-tumoral cells, we selected a GES-1 cell line that was generated from human fetal gastric epithelial cells immortalized by transformation with SV40 virus [20], in comparison with an AGS cell line that was established from fragments of a tumor, adenocarcinoma, resected from a patient who had received no prior therapy [21]. Finally, based on the transcriptomic landscape, we dissected the molecular pathways underlying the early response to *H. pylori* infection in GES-1 cells with respect to a prototypical proinflammatory signal such as TNF.

## 2. Results

### 2.1. Response of AGS and GES-1 Cells to Inflammatory Mediators Linked to Gastric Diseases

Since a thorough and comprehensive characterization of GES-1 cells has not been reported so far, we first decided to analyze the inflammatory response in this model in comparison to the widely used AGS cell line. We selected TNF, a well-established pro-inflammatory mediator linked to epithelial inflammation, including *H. pylori* infection [22]. We evaluated both the release of cytokines in both cell lines and the expression of key genes mediating the inflammatory response after 6 h treatment with TNF. In basal conditions, GES-1 and AGS cells released very low levels of inflammatory factors; accordingly, the expression of some cytokines and related factors was barely detectable or even below the detection limits (data not shown). As expected, TNF induced the release of IL-8 in both GES-1 and AGS cells (Figure 1A,B). Moreover, in GES-1 cells TNF induced IL-6 (Figure 1C) and MMP-9 release (Appendix A). In contrast, AGS cells did not secrete detectable amounts of IL-6 (Figure 1D) and MMP-9 (Appendix A).

To obtain a more comprehensive view of the inflammatory response triggered by TNF in GES-1 in comparison to AGS cells, we analyzed the expression levels of a panel of genes related to inflammation by using the Human Inflammatory Cytokines & Receptors RT^2^ Profiler PCR Array. In addition, we also evaluated the mRNA levels of *IL6* and *MMP9*, genes that were not included in the commercial array. As specified in the Material and Methods section, the analysis was performed on triplicate samples per condition. Only genes detected in all three replicates were considered for subsequent analysis, setting the cycle threshold (Ct) cut-off at 35 and normalizing versus the housekeeping gene *RPLP0* for calculation of ΔΔCt. A *t*-test employing the False Discovery Rate (FDR) method was conducted. The most relevant data of the differentially expressed genes (i.e., fold change and nominal *p*-value), are listed in Table 1. A subset of seven genes comprising chemokines and interleukins, was induced in both cell lines. The top upregulated genes were *CCL20*, *LTB*, *TNF*, and *CXCL8*. The remaining 18 genes were instead induced in GES-1 cells only, among which included *CCL2*, *MMP9*, *IL6*, and *IL1B*. The data obtained at the transcriptomic level are consistent with the results related to cytokine release. In fact, TNF treatment induced mRNA levels and secretion of IL-8 in both cell types; it also stimulated the expression and the release of IL-6 and MMP-9 in GES-1 cells but failed to do so in the AGS line.

From the mechanistic point of view, TNF induced the canonical NF-κB signaling in both cell lines, as demonstrated by significant p65 phosphorylation (Figure 2A,B) and nuclear translocation (Figure 2C), which resulted in potent stimulation of NF-κB driven transcription, as assessed by using either the NF-κB-luc construct (Figure 2D,F) or the native IL-8 promoter bearing a κB site (Figure 2E,G).

Based on this evidence we re-analyzed the results obtained from the PCR array. Indeed, of the 86 genes that were evaluated, 28 could be classified as NF-κB targets according to the Thomas Gilmore’s lab list [23] and the NF-κB target gene sets [24]. As reported in Table 1, a set of NF-κB target genes was upregulated by TNF treatment in both GES-1 and AGS cells, i.e., *CXCL8* (the gene encoding IL-8), *CCL20*, *CXCL2*, *LTB*, *CXCL5*, and *TNF*. In contrast, another group of genes, *CCL2* (the gene encoding monocyte chemoattractant protein 1), *CXCL3*, *CXCL6*, *IL15*, *CXCL1*, *CXCL10*, *IL1B*, *IL6*, and *MMP9*, was upregulated upon TNF treatment in GES-1 cells only.

Collectively, these results indicate that, although the NF-κB signaling is active in both cell lines, the inflammatory response triggered by TNF does not completely overlap in the two cell lines. The different response could be ascribed to TNF-regulated expression and/or activation of factors others than members of the Rel family, operating in GES-1 cells and less functional in AGS cells.

The NF-κB signaling pathway is also regulated through ubiquitinylation of several adapter proteins and regulators [25,26]. While interfering with protein ubiquitinylation attenuates NF-κB-mediated transcription, proteasomal inhibition with the MG132 compound induces *CXCL8* expression via adaptor protein-1 (AP-1) activation [25,26]. Therefore, we explored the interplay between proteasome-dependent protein degradation, TNF-induced NF-κB signaling and the pattern of inflammatory markers by using the proteasome inhibitor MG132.

In GES-1 and AGS cells treated with MG132 the NF-κB-driven transcription (Figure 3A,B) and IL-8 promoter activity (Figure 3C,D) induced by TNF were significantly reduced, probably because of decreased degradation of the NF-κB inhibitor I-κBα, as reported in the literature. In line with this observation, the secretion of IL-8 was not or was less induced after 3 h exposure to TNF in the presence of MG132 in both cell lines (Figure 3E,F, left panels). However, a different response was observed at longer times. In fact, at 6 h, inhibition of protein ubiquitinylation was able per se to increase IL-8 secretion in GES-1 cells (Figure 3E, right panel); moreover, the secretion of IL-8 by GES-1 cells co-treated with TNF and MG132 was similar to that observed in the presence of TNF alone, despite the attenuated NF-κB-driven transcription and IL-8 proximal promoter activity discussed above. Of note, in AGS cells, also at 6 h TNF-induced secretion of IL-8 was blunted in the presence of proteasomal inhibition (Figure 3F, right panel). mRNA levels of IL-8 measured at 6 h (Figure 3G,H) mirrored the secretion data in both GES-1 and AGS cells. All together, these data suggest that in conditions of inhibition of protein ubiquitinylation, (i) the NF-κB pathway initiated by TNF is less functional in both cell lines, and (ii) at longer times (6 h) other factors operating in GES-1 cells but not in AGS cells are able to sustain IL-8 expression and secretion.

### 2.2. H. pylori-Induced Inflammatory Response in GES-1 and AGS Cells

To investigate the effects of *H. pylori* infection in GES-1 and AGS cells, we used the reference strain 26,695 from ATCC (ATCC 700392^TM^) expressing the virulence factor CagA and the most virulent allelic isoforms of the virulence factor VacA S1/M1, thereafter named reference strain 26,695 or ATCC strain [27]. In parallel, we also used a clinical isolate (strain #6), which is CagA-negative and bears the less virulent isoforms of VacA S2/M2. The genetic features and antibiotic resistance of the two strains are reported in Appendix A. To assess the ability of the pathogens to interact with the host cells, we performed an adhesion test with FITC-labeled bacteria. As shown in Figure 4A, both *H. pylori* strains interacted similarly with the two cell types (mean % of adherent over total bacteria: 2.73 and 3.43 to GES-1 cells and 2.12 and 3.05 to AGS cells), thus indicating that the virulence potential of the two strains has negligible effects on their ability to interact with the host. Furthermore, the two gastric cell lines most likely express similar membrane-associated systems mediating *H. pylori* adhesion since the fraction of adherent bacteria was similar in AGS and GES-1 cells, being 2.73% (GES-1 cells) and 2.12% (AGS cells) for the ATCC strain, and 3.43% (GES-1 cells) versus 3.05% (AGS cells) for the clinical isolate #6.

Next, we evaluated the release of IL-8 and IL-6 upon infection of GES-1 and AGS cells with the two strains. In line with previous studies [28], the CagA-negative strain (clinical isolate #6) induced IL-8 secretion to a lesser extent with respect to the ATCC CagA-positive strain and the difference was observed in both cell types (Figure 4B,C). In contrast, IL-6 release was induced in GES-1 cells only, independently of the virulence potential of the *H. pylori* strains (Figure 4D,E). The data on IL-6 secretion, in line with those obtained upon treatment with TNF, support the hypothesis that some key factors required for IL-6 expression and secretion were not activated in AGS cells in our setting.

To assess the involvement of NF-κB signaling in the host response to *H. pylori*, we evaluated the phosphorylation state and nuclear translocation of p65. As shown in Figure 5 (panel A–D), only the CagA-positive strain was able to induce phosphorylation of p65 in GES-1 and AGS cells. Of note, the maximal degree of phosphorylation was obtained in both cell lines at later times after infection with the ATCC strain (60 min) with respect to TNF treatment (30 min) (see Figure 2A,B for comparison). These data are in line with previous studies reporting that the activation of the NF-κB pathway by *H. pylori* follows a slower kinetics with respect to TNF [29]. Unfortunately, we were unable to evaluate p65 translocation to the nuclear compartment of infected GES-1 and AGS cells with ELISA; therefore, we assessed the intracellular distribution of p65 under basal and stimulated conditions with confocal microscopy. As shown in Figure 5E,F, middle and bottom panels, 1 h-treatment with TNF clearly induced nuclear translocation of p65 in both GES-1 and AGS cells, as expected. In addition, infection with the ATCC strain induced non-homogeneous nuclear translocation of p65 in GES-1 cells (Figure 5E), consistent with a lesser degree of p65 phosphorylation with respect to TNF, as assessed with western blotting (Figure 5A). A similar profile was observed in AGS cells (Figure 5F, middle and bottom panels). On the other hand, infection with the CagA-negative strain (clinical isolate #6) did not induce p65 translocation in the nuclear compartment of both GES-1 and AGS cells (Figure 5E,F, rightmost panels). These data are consistent with the observation that in cells infected with this strain the phosphorylation state of p65 was not significantly changed with respect to control cells (Figure 5C,D).

Since *H. pylori* has been reported to activate the non-canonical NF-κB pathway [30], we also evaluated the processing of p100 to p52/RelB as a readout of this signaling cascade. Up to 2 h after infection with the ATCC strain, no formation of p52 was observed (data not shown), thus ruling out the possibility that the phenotypic effects observed in our setting upon *H. pylori* infection could be due to activation of the alternative NF-κB route.

We also investigated the effect of *H. pylori* infection on the expression of the same panel of inflammatory genes analyzed in TNF-treated cells. As summarized in the Appendix A, the expression profiles obtained from this assay were not consistent with the data on cytokine secretion reported in Figure 4. Especially, we found lack of statistically significant induction of *CXCL8* and *IL6* in GES-1 and of *CXCL8* in AGS cells. After thorough evaluation of the results, we ascribe these discrepancies to multiple factors, including basal expression levels of the analyzed genes and biological variability associated with bacterial infection that may have weakened the use of the PCR array to capture transcriptomic signatures in this specific experimental setting.

As described above for TNF, we also evaluated the potential role of proteasome in the host response to *H. pylori* infection by using the MG132 inhibitor. After 3 h infection, the *H. pylori*-induced secretion of IL-8 was reduced by co-treatment with MG132 in both GES-1 and AGS cells (Figure 6A and Figure 6B, respectively). Of note, the effect of proteasomal inhibition was seen independently of the *H. pylori* strain, thus independently of the activation of the NF-κB pathway. After 6 h infection with *H. pylori*, the secretion of IL-8 was increased as expected, and in parallel the expression of *CXCL8* was also induced several times in both GES-1 and AGS cells (Figure 6C and Figure 6D, respectively). Treatment of *H. pylori* infected GES-1 cells with the MG132 compound modestly affected the ATCC strain-induced secretion of IL-8; moreover, it seemed to potentiate the cytokine release in cells infected with the clinical isolate strain #6 (Figure 6A) even though data did not reach statistical significance. A similar trend was also observed at the mRNA level (Figure 6C). In contrast, in *H. pylori*-infected AGS cells, proteasomal inhibition inhibited IL-8 secretion and expression (Figure 6B,D, respectively), especially in ATCC-infected cells. Due to high dispersion of fold change, data did not reach statistical significance in AGS cells infected with the clinical isolate #6.

Collectively, our results on the NF-κB pathway in *H. pylori* infected cells demonstrate a mild activation by the CagA-positive strain in both GES-1 and AGS cells based on p65 phosphorylation and nuclear translocation. Unfortunately, through the analysis of the PCR array we were unable to demonstrate that activation of the NF-κB pathway leads to a significant and consistent upregulation of its target genes, including *CXCL8* and *IL6*, probably due to intrinsic limitations of the approach, as mentioned above.

The fact that both strains induced IL-8 (Figure 4B) and IL-6 (Figure 4D) secretion by GES-1 cells, independently of CagA and p65 phosphorylation, suggests that transcription factors others than members of the NF-κB family and other pathways might be involved in the host response to *H. pylori* in this cell line. This hypothesis is indirectly confirmed by the results obtained using the proteasomal inhibitor MG132, which suggest that in GES-1 cells IL-8 expression and secretion is also supported by pathways that affect gene transcription and are sensitive to or involved in proteasomal-dependent protein degradation.

### 2.3. Transcriptome Analysis of H. pylori Infection in GES-1 Cells

The results shown so far indicate that while the response to inflammatory stimuli and pathogens in AGS cells mainly relies on activation of NF-κB signaling, the response observed in GES-1 cells involves additional pathways and is probably closer to gastric pathophysiology. Thus, our next objective was to gain further insight into the molecular pathways triggered in the early stages of *H. pylori* infection in GES-1 cells by performing a transcriptomic analysis. We used the ATCC *H. pylori* strain that, according to the results shown above, activated both the NF-κB signaling cascade and other pathways underlying the host response and pathogenetic mechanisms. To dissect the pathogen-specific transcriptomic signature, we performed in parallel the analysis of GES-1 cells treated with TNF. In fact, by comparing the transcriptomic fingerprint obtained with the two stimuli, we expected to identify those more specifically associated to *H. pylori* infection with respect to those associated to more general mechanisms of inflammation.

RNA was collected and sequenced from GES-1 cells treated with TNF or infected with *H. pylori*, reference strain 26695, (each condition in three independent replicates). As a control, noninfected/nontreated GES-1 cells were used (three independent replicates) The principal component analysis (PCA) shown in Figure 7 shows the strong clustering of these samples according to their phenotype.

Treatment with TNF profoundly affected the transcriptome of GES-1 cells as shown in Figure 8A,B. A total of 1848 genes were differentially expressed, of which, 961 were upregulated and 897 downregulated. Differentially expressed genes (DEGs) were used to perform an enrichment analysis to evaluate which pathways were over-represented. The analysis was carried out using EnrichR and referred to the GO_Biological_Process_2021 database. The top pathways for both the upregulated and downregulated genes are reported in Table 2 while the full lists can be found in Appendix A. As, expected, among the most significant upregulated pathways we found the cytokine-mediated signaling pathway, cellular response to cytokine stimulus, and regulation of inflammatory response. Instead, downregulated DEGs clustered in Gene Ontology (GO) terms linked to cell specification and differentiation, mainly neurons and neural cell.

The transcriptomic changes induced by *H. pylori* infection were even more pronounced; in fact, we found 2072 upregulated and 1501 downregulated genes, for a total of 3573 DEGs (Figure 9A,B). Of note, *CXCL8* was the most upregulated gene; moreover, in the list of upregulated DEGs we also found *ICAM1*, *BIRC3*, *CCL20*, *TNFAIP2*, and *NFKBIA*, transcripts that were identified as a primary response in human gastric organoids infected with *H. pylori* [10]. However, since the same subset of genes were upregulated by TNF they could not be considered specific hallmarks of *H. pylori* infection. Pathway analysis revealed that the most represented GO terms in genes upregulated by *H. pylori* infection were mainly related to regulation of transcription, but also to cytokine-related pathways, apoptosis, angiogenesis, and cell proliferation (Table 3 and Appendix A). On the other hand, the most represented GO terms in *H. pylori* downregulated genes were linked to protein folding and response to unfolded proteins (Table 3 and Appendix A).

We then compared the transcriptomic profiles obtained in TNF-treated and *H. pylori* infected cells in comparison to untreated cells. As shown in the heatmap (Appendix A), a different trend in gene expression was found upon TNF treatment with respect to *H. pylori* infection, as well as in comparison with untreated cells. Of note, both TNF treatment and *H. pylori* infection regulated a set of common genes (886 DEGs), of which 508 were upregulated by both treatments, 326 downregulated by both treatments, and 48 displayed a different trend in the two groups. As expected, the common upregulated genes clustered in GO terms such as cytokine-mediated signaling pathway, regulation of inflammatory response, and regulation of NIK/NF-κB signaling (Appendix A).

To further identify a specific transcriptomic signature ascribable to *H. pylori* infection and distinguishable from that induced by TNF treatment, we compared the RNA-seq data obtained from the two sets of samples. As shown in the heatmap (Figure 10A), a different trend in gene expression between the two groups can be clearly observed. A total of 3533 DEGs were identified, with 2065 upregulated and 1468 downregulated genes in cells infected with *H. pylori* compared to cells treated with TNF (Figure 10B). The enrichment analysis, shown in Table 4 and Appendix A, revealed several pathways with upregulated genes significantly linked to *H. pylori* infection such as regulation of gene transcription, apoptosis, and angiogenesis. The specific fingerprint of *H. pylori* on the transcription machinery of infected GES-1 cells is mostly linked to upregulation of members of the zinc finger proteins (ZNFs) family of transcription factors.

Among the most enriched pathways with downregulated genes resulting from the comparison of *H. pylori* infected vs. TNF-treated cells, we found cytokine-mediated signaling pathway and positive regulation of I-κB kinase/NF-κB signaling. This finding is most likely because the expression of genes clustering in these GOs was more affected by TNF treatment than by *H. pylori* infection. This is the case, for example, of *TNF*, *CSF1*, *BIRC3*, *LTB*, *CCL2*, *CXCL5*, and *CD40*.

## 3. Discussion

Gastritis caused by *H. pylori* infection is still highly diffused worldwide, in all geographic areas, thus maintaining the elevated need for studies on pathogenetic mechanisms, target identification, and new therapeutic agents or strategies. Because of the high turnover rate of gastric epithelial cells in a stomach colonized by *H. pylori*, populations of cells at different stages of infection coexist [17]. Consequently, from the mechanistic point of view, early pathogenetic pathways triggered by *H. pylori* are activated at any time, in the fraction of newly differentiated cells, together with chronically established responses in more aged cells. Organoids generated from primary gastric cells represent a powerful system to study the host response to *H. pylori* infection [10]. However, gastric epithelial cell lines remain a robust model system in the field either to validate molecular pathogenetic pathways or for screening purposes.

On these premises, we investigated the early host response to *H. pylori* in a cell line of gastric epithelial cells that has not been fully characterized before and exhibits the advantage of being of non-tumoral origin, in contrast to the cell lines that are commonly used.

Despite that several markers of inflammation and carcinogenesis have been investigated in GES-1 [15,16], our results provide an in-depth characterization of the inflammatory response of GES-1 cells in comparison with the widely used AGS cells derived from a stomach tumor. In parallel to infection with *H. pylori*, we also examined the response to TNF, a prototypical cytokine characterizing virtually all inflammatory conditions, including *H. pylori*-induced gastritis [31]. By evaluating the expression and secretion of inflammatory factors, and the activation state of NF-κB signaling we found that GES-1 cells provide a more complete response in comparison to AGS, a response that more closely resembles gastric pathophysiology.

By analyzing the activation state of NF-κB signaling, cytokine expression, and secretion in the presence of a proteasomal inhibitor, we found that while in our setting NF-κB signaling is the prevalent pathway underlying the response of AGS to TNF and CagA-positive *H. pylori* infection, additional pathways operate in GES-1 cells. One of the consequences of this difference is that, while activation of NF-κB is sufficient to upregulate expression of the *CXCL8* gene in AGS cells, the same does not apply to *IL6*, whose induction requires the activation of multiple factors and pathways. The CCAAT/enhancer-binding protein family (CEBP) includes several members acting as transcription factors and IL6 is one of the main target genes transactivated by these factors [32]. The transcriptomic analysis of GES-1 cells showed that both *H. pylori* infection and treatment with TNF induced the expression of genes belonging to this family, which likely contributed to the upregulation of *IL6* in this cell model, in combination with the action of factors of the NF-κB pathway. The contribution of multiple signaling pathways, including CEBP proteins, in the host response would be in line with the data that we obtained in the presence of the proteasomal inhibitor MG132. In fact, in conditions in which NF-κB signaling is inhibited (MG132), TNF treatment or *H. pylori* infection still upregulated *CXCL8* and *IL6* in GES-1 cells. A further indication is provided by the fact that infection with the CagA negative strain (clinical isolate #6) induced IL6 expression in GES-1 cells independently of NF-κB activation.

The transcriptomic profile further unveiled that potentially CEBP proteins play a more specific role in *H. pylori* infection. In fact, *H. pylori* infection but not treatment with TNF induced the expression of genes associated with CEBPβ, such as the oncogene Myc, Tenascin C, and Thrombospondin 1 that are involved in extracellular matrix remodeling [33].

Moreover, our data on the transcriptomic profile indicated that upregulation of numerous genes clustering in the GO terms, encoding zinc-finger proteins (ZNFs), regulation of transcription, DNA-templated (GO:0006355), and regulation of transcription by RNA polymerase II (GO:0006357) (Figure 9C) is a specific trait of *H. pylori* infection. These factors are involved in biological processes such as development and differentiation but also tumorigenesis and tumor progression [34]. The involvement of ZNFs in gastric cancer has been recently reviewed [35]. Some of the ZFs modulated in gastric cancer with a proven role in the related processes, were regulated in the same manner, upregulation, or downregulation, in GES-1 cells infected with *H. pylori*. ZNFs playing a role in inflammation (i.e., ZHX2), cell proliferation, epithelial to mesenchymal transition, invasion and metastasis (i.e., ZC3H15, ZBTB20, RNF114, ZEB2), drug resistance and cell cycle (i.e., GLIS2), apoptosis, epithelial to mesenchymal transition, and invasion and metastasis (i.e., RNF43) were up- or downregulated as early as 6 h after infection with *H. pylori* in comparison to non-infected cells in our experimental setting. On the other hand, some ZNFs were oppositely regulated in GES-1 infected cells with respect to what was observed in gastric cancer; this was the case, for example, for KLF4, KLF6, KLF9, GLI2, and TNFAIP3, which participate in cellular processes such as cell cycle, cell proliferation, epithelial to mesenchymal transition, invasion and metastasis, apoptosis, and inflammation. The analysis of the transcriptomic profile of TNF-treated cells showed that most of the ZNFs regulated by *H. pylori* were also modulated by TNF. Collectively, these findings indicate that specific ZNFs represent early targets of *H. pylori* or inflammatory cues, with potential consequences on the activation/modulation of the tumorigenic program. Nevertheless, the role of other ZNFs that are specifically upregulated by *H. pylori* needs to be further clarified. Of note, both *H. pylori* infection and TNF treatment induced the expression of members of the metallothionein family, proteins that are involved in the distribution and cellular utilization of dietary zinc [36], therefore linked to zinc finger-containing proteins.

It is known that *H. pylori* infection leads to activation of the activator protein 1 (AP-1) cascade [37] and to increased expression of members of this family of transcription factors [38]. Our transcriptomic analysis revealed that in *H. pylori*-infected GES-1 cells, the expression of AP-1 related genes such as *FOS*, *FOSB*, *FOSL1*, *FOSL2*, and *JUN*, was statistically increased with respect to both non-infected and TNF-treated cells. Of note, *DKK1*, the gene encoding the Dickkopf-related protein, which was recently identified as a target of *H. pylori* activated AP-1 signaling [38], was upregulated as well in infected GES-1 cells in comparison to either non-infected or TNF-treated cells. Thus, it may be concluded that AP-1 signaling is operating in *H. pylori* infected GES-1 cells also due to direct increased expression of this set of transcription factors. This could ultimately lead to upregulation of specific downstream target(s) that may be considered hallmarks of *H. pylori* infection and contribute to its pathogenicity. Consequently, not only agents able to impair AP-1 signaling but also to decrease the expression levels of these factors could be beneficial in counteracting *H. pylori* effects.

The analysis of the enriched downregulated pathways emerged from the transcriptomic profile of *H. pylori* infected GES-1 cells provided further insights on the cellular processes that are modulated by the bacterium. In fact, among the most downregulated pathways we found GO terms related to protein folding of de novo synthesized proteins and response to unfolded proteins; the downregulated genes listed in these sets encode several molecular chaperones and related proteins. The role of these proteins, and, in general, of the unfolded protein response (UPR), in gastric pathophysiology needs to be investigated in detail, considering the possibility that after an initial phase of attenuation [39], induction of UPR may follow at the stage of *H. pylori* positive gastric cancer, as previously reported [40]. Indeed, *H. pylori* infection has been shown to affect multiple aspects of cell biology, among which are cell death and endoplasmic reticulum stress; however, in *H. pylori* positive tumor lesions, adaptive responses leading to increased survival of malignant cells may also occur [41], thus making it difficult to establish the exact contribution of these pathways to *H. pylori* pathogenicity.

In our study, in *H. pylori*-infected vs. TNF-treated GES-1 cells we also found downregulation of genes clustering in the GO terms NADH dehydrogenase complex assembly and mitochondrial respiratory chain complex assembly. This finding would suggest that altered levels of some subunits of complex I may be observed upon *H. pylori* infection, nevertheless, no significant changes in the expression levels of these genes were observed comparing *H. pylori*-infected vs. non-infected GES-1 cells. In this regard, it is worth noting that *H. pylori* induced mitochondrial alterations have been recently reported in GES-1 cells [42] even though these effects, potentially leading to mitophagy, are ascribable to the well documented organelle damaging activity of vacuolating cytotoxin A (VacA). Thus, we believe that potential effects on the expression of complex I subunits due to *H. pylori* would be less relevant with respect to the direct effects of VacA on organelles’ structure and mitochondrial functions.

In conclusion, our study provides an in-depth characterization of the early response to the prototypical cytokine TNF and to *H. pylori* in a non-tumoral cell model. By complementing different biological assays and bioinformatics analysis of transcriptomic data, we were able to shed light on transcriptional circuits and related cellular processes that play a role in *H. pylori* pathogenicity and could be explored as pharmacological targets.

## 4. Materials and Methods

### 4.1. Reagents

RPMI 1640 medium, DMEM F-12, penicillin, streptomycin, L-glutamine, and trypsin-EDTA were purchased from Gibco (Life Technologies Italia, Monza, Italy). Fetal bovine serum (FBS) and disposable materials for cell culture were purchased from Euroclone (Euroclone S.p.A., Pero-Milan, Italy). SV-40 immortalized GES-1 from human gastric epithelium were kindly donated by Dr. Dawit Kidane-Mulat (University of Texas at Austin), while human adenocarcinoma cells AGS (ATCC, CRL-1739^TM^) were from LGC Standard S.r.l., Milan, Italy.

Mueller Hinton Broth, Brucella Broth, and glycerol from BD (BD, Franklin Lakes, USA), agar from Merck Life Science (Merck Life Science S.r.l., Milan, Italy), and defibrinated sheep blood from Thermo Fischer Scientific (Oxoid^TM^, Hampshire, UK) were used to cultivate and store *H. pylori*, CagA-positive strain 26,695 from ATCC (ATCC 700392^TM^, Virginia, USA) and clinical strain #6 (Sant’Orsola Hospital, Bologna, Italy).

The reagent 3-(4,5-dimethylthiazol-2-yl)-2,5-diphenyltetrazolium bromide (MTT), fluorescein-5-isothiocyanate (FITC), the proteasome inhibitor MG132, and the Protease Inhibitor Cocktail were purchased from Merck Life Science (Merck Life Science S.r.l., Milan, Italy). Lipofectamine^®^ 3000 and Carboxyfluorescein succinimidyl ester (CFSE) 5 mM (CellTrace^TM^, Cell Proliferation kits) and the ActinRed^TM^ 555 ReadyProbes^TM^ reagent were from Invitrogen (Thermo Fisher Scientific, Waltham, MA, USA). The Britelite^TM^ Plus reagent was from Perkin Elmer (Milan, Italy).

Rabbit antibodies for p65, p100, phospho-p65, the secondary anti-rabbit antibody conjugated with Alexa Fluor 647, and ProLong Gold Antifade Reagent with 4′,6-diamidino-2-phenylindole (DAPI) (#8961) were purchased from Cell Signaling Technology (Danvers, MA, USA); mouse anti-β-actin and the secondary anti-mouse HRP conjugate were from Merck Life Science (Merck Life Science S.r.l., Milan, Italy). The Human TNF and Human IL-8 or IL-6 ABTS Elisa Development Kit were from Peprotech Inc. (Peprotech Inc., London, UK). The Human MMP-9 Elisa Development Kit was from RayBiotech Life (RayBiotech Life, Inc. 3607 Parkway Lane Suite 200, Norcross, GA, USA). The Human Inflammatory Cytokines & Receptors RT^2^ Profiler PCR Array (PAHS-011ZE) and miRNeasy Mini Kit were purchased from QIAGEN (Milan, Italy). The BCA protein assay kit was from Euroclone (Euroclone S.p.A., Pero-Milan, Italy).

### 4.2. Cell Culture and Treatments

Human gastric epithelial cell lines of non-pathologic (GES-1) or tumoral nature (AGS, from gastric adenocarcinoma) were cultivated, respectively, in RPMI 1640 and DMEM F-12 medium, supplemented with penicillin 100 units/mL, streptomycin 100 mg/mL, 1% L-glutamine 2 mM, and 10% heat-inactivated FBS. Cells were incubated at 37 °C, 5% CO_2_, in a humidified atmosphere. For the subculture, cells were detached by trypsin ethylenediamine-tetra-acetic acid (EDTA) 0.25% solution and cultivated in new flasks with fresh medium (1 × 10^6^ cells) every 48–72 h. The early inflammatory response was induced by TNF (10 ng/mL) or *H. pylori* infection (ratio cell:bacteria 1:50), using FBS-free medium, for the time indicated for each experiment (from 30 min to 6 h).

When specified, the proteasome was inhibited by treating cells with MG132 (10 μM) 1 h before and during the pro-inflammatory challenge.

### 4.3. Cytokine Assays

Cells were seeded at the density of 3 × 10^4^ cells/well (24-well plate) for 48 h then treated with TNF or *H. pylori* for 3 h and 6 h. IL-8 and IL-6 were quantified in 100 μL of cell medium, after at least three independent experiments, with an ELISA assay. The assay was conducted according to the manufacturer’s instructions, as previously reported [11]. At the end of the assay, the absorbance of the samples at 405 nm was compared to the absorbance of a calibration curve made with human recombinant standard IL-8 (0–1000 pg/mL). The quantitative release of inflammatory mediators in each sample was expressed as mean (pg/mL) ± SEM and compared to the unstimulated control.

### 4.4. H. pylori Strains Infection and Gastric Cell Infection

*H. pylori* 26,695 and #6 strains were characterized for virulence and antibiotic resistance during the present study as reported in the Results session. Both strains were cultured in agar-blood Petri dishes, as previously mentioned [11]. Briefly, in plates containing Mueller–Hinton Broth medium, 5% agar, and 25% blood, the bacteria were incubated for 72 h, under a microaerophilic atmosphere (5% O_2_, 10% CO_2_, and 85% N_2_) at 37 °C, 100% humidity. Then, the bacteria were collected and counted by optical density (OD) at 600 nm. Since the OD value of 5 corresponds to 2 × 10^8^ bacteria, the bacterial suspension was specifically adjusted to infect cells with bacterium-to-cell ratio 50:1, for 1 h or 6 h, depending on the biological assay. The treatments were conducted with serum- and antibiotic-free medium. To allow the synchronization of cells, serum starvation was performed the day before treatment, using 0.5% FBS medium, supplemented with 1% L-glutamine and 1% penicillin/streptomycin. During the treatment, infected cells were maintained in aerobic atmosphere, at 37 °C and 5% CO_2_.

### 4.5. Bacterial Adhesion to Gastric Cells

The adhesion of *H. pylori* to gastric cells was measured with a fluorometric method adapted from Messing and colleagues [43]. AGS and GES-1 cells were seeded in 96-well plates for 48 h (1 × 10^4^ cells) before the infection with FITC-labeled *H. pylori*. FITC (2 μL) solution (1% in DMSO) was added to 10^8^ bacteria suspended in PBS 1X and incubated for 45 min at 37 °C; then, the bacterial suspension was centrifuged (3150× *g*, 5 min) and washed twice (PBS 1X) to remove the excess of probe and resuspended in PBS 1X for cells infection. After 1 h cells were washed twice with PBS 1X, 100 μL of PBS 1X was added, and the fluorescence was quantified at excitation/emission of 485/535 nm.

### 4.6. Gene Expression

The expression of 84 selected genes, known to regulate the inflammatory response, was evaluated with rt-PCR array (Human Inflammatory Cytokines & Receptors RT^2^ Profiler PCR Array), as previously described [44]. In this array, each well contained the primers for a specific target gene or housekeeping gene. A diluted aliquot of cDNA, equivalent to 400 ng of total RNA, was mixed with the SYBR Green Master Mix RT^2^ reagent (QIAGEN) according to the manufacturer’s instructions and loaded into the 384-well array. The real-time PCR was performed using the CFX384^TM^ Real-Time PCR Detection System (coupled to C1000^TM^ Thermal Cycler; Bio-Rad Laboratories S.r.l., Segrate, Italy). The threshold cycle value for each gene (Ct) was automatically provided by the management software CFX Manager^TM^ (Bio-Rad Laboratories S.r.l., Segrate, Italy), depending on the amplification curves. The baseline and the threshold values were set manually as recommended by the PCR array manual. The analysis of the data was performed using the web portal SABiosciences (QIAGEN Sciences, Germantown, MD, USA). Each specimen underwent triplicate testing, resulting in six samples analyzed for both the TNF and *H. pylori* treatments (comprising 3 treated samples and 3 control samples). Only genes detected in all three replicates were considered for subsequent analysis. The cycle threshold (Ct) cut-off was established at 35, and genes with Ct values of 35 or lower were deemed not expressed and consequently excluded from further analysis. To normalize the data, the housekeeping gene RPLP0 was employed, enabling the calculation of ΔCt, denoting the disparity between the Ct value and the mean expression of the RPLP0 gene in either the case or control samples. Subsequently, a *t*-test employing the False Discovery Rate (FDR) method was conducted through an R routine, leading to the determination of ΔΔCt for each gene. This parameter represents the difference between the mean expression levels among cases and controls. Finally, the Fold Change (FC) was computed as FC = 2^−ΔΔCt^ to quantify the alteration in gene expression between the case and control groups.

The modulation of *IL8* and *IL6* gene expression, specifically, was also evaluated with the TaqMan^TM^ method. Sequences of the primers and probe sets are available upon request. The PCR was performed on 10 ng mRNA/well in the Real-Time System Bio-Rad CFX384 (Bio-Rad, Hercules, CA, USA) using an iTaq^TM^ Universal (Bio-Rad) and probes, which allowed mRNA reverse transcription and cDNA labelling. The thermal cycling protocol required a preliminary step for reverse transcription at 50 °C for 10 min, followed by polymerase activation step (95 °C, 5 min) and 40 cycles with 95 °C denaturation step for 10 s and annealing/extension at 60 °C for 30 s. All the samples were tested in triplicate, and the relative expression of genes was calculated by normalizing the threshold cycle (Ct) of each gene with the Ct of GAPDH mRNA to correct for variation in RNA loading.

### 4.7. Promoter Activity Assays

AGS and GES-1 cells were seeded in 24-well plates for 48 h (3 × 10^4^ cells), then transiently transfected with reporter plasmids, as previously described [14]. Plasmids included the luciferase gene under the control of the E-selectin promoter (NF-κB-Luc) or IL-8 promoter (IL-8-Luc), containing κB elements responsive to NF-κB (50 ng per well). Lipofectamine^®^ 3000 reagent was used for the transfection assays, according to manufacturer’s instructions. NF-κB-Luc plasmid was a gift from Dr. N. Marx (Department of Internal Medicine-Cardiology, University of Ulm; Ulm, Germany), while IL-8-Luc plasmid was provided by Dr. T. Shimohata and Prof. A. Takahashi (University of Takushima, Japan). The day after, cells were treated with TNF for 6 h. At the end of the treatment, luciferase activity into cells was measured using the Britelite^TM^ Plus reagent, according to the manufacturer’s instructions. Results (mean ± SEM of at least three experiments) were expressed as the percentage relative to the unstimulated control, to which a value of 100% was arbitrarily assigned.

### 4.8. NF-κB Pathway

The p65 phosphorylation and p100 processing was analyzed using western blot, while nuclear translocation of NF-κB was analyzed with an ELISA assay and immunofluorescence techniques, as previously reported [11].

#### 4.8.1. Western Blot Analysis

Total protein extracts from gastric epithelial cells were obtained by lysing cells with 200 μL radioimmunoprecipitation assay (RIPA) buffer containing a mix of protease and phosphatase inhibitors (1 mM sodium orthovanadate (Na_3_VO_4_) and 5 mM sodium fluoride (NaF). The protein concentration for each cell lysate was assessed with the bicinchoninic acid (BCA) protein assay method, according to manufacturer instructions, and 20 μg of each sample was prepared with a 4X Laemmli sample buffer, boiled at 95 °C for 5 min, centrifuged at 16,000× *g* for 1 min, and loaded on sodium dodecyl-sulfate (SDS)-polyacrilamide gel. After gel run, proteins were transferred to a nitrocellulose membrane and blocked in tris buffered saline with Tween (TBS-T) containing 5% BSA for 1 h at room temperature. Membranes were incubated overnight at 4 °C with the primary antibodies anti-phospho-p65 (Phospho NF-κB p65 (Ser536) (93H1) Rabbit mAb #3033; Cell Signaling Technology, Danvers, MA, USA) and anti-β-actin (Monoclonal Anti-β-Actin Clone AC-15 produced in mouse; Merck Life Science, Milan, Italy), used as a control. Then, after washing with TBS-T solution, membranes were incubated with anti-rabbit and anti-mouse secondary antibodies (Merck Life Science, Milan, Italy), respectively, for 1.5 h at room temperature. Immunocomplexes were visualized with electrochemiluminescence (ECL) (Westar Antares, Cyanagen S.r.l., Bologna, Italy) and the Chemidoc MP imaging system (Bio-Rad Laboratories S.r.l., Segrate, Italy). After visualization, the membrane was stripped with a Stripping Buffer Solution (Cyanagen S.r.l., Bologna, Italy) and reprobed with the primary antibodies anti-p65 (NF-κB p65 (D14E12) XP^®^ Rabbit mAb #8242; Cell Signaling Technology, Danvers, MA, USA), then with the anti-rabbit secondary antibody (Merck Life Science, Milan, Italy) and visualized as described above. Protein levels were quantified with ImageLab 6.1 software (Bio-Rad Laboratories S.r.l., Segrate, Italy). Primary antibodies were diluted as follows: anti-phospho-NF-κB(p65), p65, p100, 1:1000 *v*/*v*; anti-β-actin 1:2500 *v*/*v*.

#### 4.8.2. Immunofluorescence Analysis

The immunofluorescence technique was used to assess the translocation of NF-κB from cytoplasm to nucleus of gastric cells, challenged with TNF and *H. pylori* strains. Cells were seeded on coverslips placed in 24-well plates at the density of 3 × 10^4^ cells. Before treatment, *H. pylori* was stained with CFSE 5 mM (5 μL was added to 2 ×10^8^ bacteria suspended in PBS 1X) and incubated in the dark for 20 min at 37 °C. Subsequently, FBS was added to the bacterial suspension for 15 min at room temperature to quench the CFSE reaction, followed by three washes with PBS 1X and centrifugation at 3150× *g* for 5 min to remove the excess of CFSE not bound to the bacterium. After 1 h treatment, co-cultures were washed (PBS 1X) and fixed with 4% formaldehyde solution for 15 min at r.t. A 5% BSA blocking solution was added to the well and incubated at room temperature for 1 h. Cells were incubated with the primary antibody (NF-κB p65 (D14E12) XP^®^ Rabbit mAb #8242) diluted 1:400 *v*/*v* overnight at 4 °C and then, after three washes with PBS 1X, with the secondary antibody (anti-rabbit IgG conjugated with Alexa Fluor 647, #4414) diluted 1:1000 *v*/*v*. After 2 h, coverslips were washed three times with PBS 1X and mounted on slides with a drop of DAPI, and then were imaged with a confocal laser scanning microscope (LSM 900, Zeiss, Oberkochen, Germany).

### 4.9. RNA-Sequencing

Cells were seeded in a 12-well plate at a density of 6 × 10^4^ cells/well for 48 h and mRNA isolated using miRNeasy Mini Kit. Cells were treated for 6 h with TNF and *H. pylori* strain 26,695 from ATCC. Three replicates were used for each condition. The quality and the concentration of the mRNA were assessed with RNA Screen Tape (Agilent Technologies, Santa Clara, CA, USA). The mRNA (500 ng) was fragmented and converted into complementary DNA (cDNA) through Illumina^®^ Stranded mRNA Prep (Illumina, San Diego, CA, USA) according to the manufacturer’s instructions. The quality and concentration of the final dual-indexed libraries were checked with D1000 Screen Tape (Agilent Technologies, Santa Clara, CA, USA). The NextSeq 500/550 High-Output Kit v2.5 was used to perform the sequencing through NextSeq 550 instrument (Illumina, San Diego, CA, USA).

### 4.10. RNA-Sequencing Data Analysis

For RNA sequencing data analysis, Illumina blc2fastq software was used to generate the Fastq files. The quality of each Fastq was inspected individually using the FASTQC tool vs. 0.11.9 [45], and Multiqc vs. 1.10.1 was then used to assess the overall good sequencing quality [46]. After the quality check, reads were aligned to the human reference genome GRCh38.p13 using STAR 2.7.9a [47], while Feature Counts 2.0.1 was used to obtain the counts for each sample [48] (see Appendix A). The rlog function was applied to transform the count data, reducing differences among samples, and normalizing for the library size. Transformed data were used to perform principal component analysis (PCA) to evaluate the sample distribution and their clustering within groups representing the same condition. Normalization and differential gene expression analyses were carried out using the Bioconductor package DeSeq2 [49]. Following normalization, differentially expressed genes (DEGs) were detected. The Wald test in DESeq2 is the default test used for hypothesis testing. Genes were considered differentially expressed when they had AdjPval < 0.05 and log2FoldChange ≥ 0.58, indicating a fold change > 1.5 in either direction. The DEGs were then used to perform the enrichment analysis with the R package “EnrichR” [50]. The enriched pathways were determined using the GO_Biological_Process_2021 database.

### 4.11. Statistical Analysis

All biological results were expressed as the mean ± SEM of at least three independent experiments. Data were elaborated through an unpaired ANOVA test and Bonferroni post-hoc analysis, using GraphPad Prism 8.0 software (GraphPad Software Inc., San Diego, CA, USA). Where specified and required, unpaired *t* test instead of ANOVA was applied to compare two experimental conditions.

## Figures and Tables

**Figure 1 ijms-24-15147-f001:**
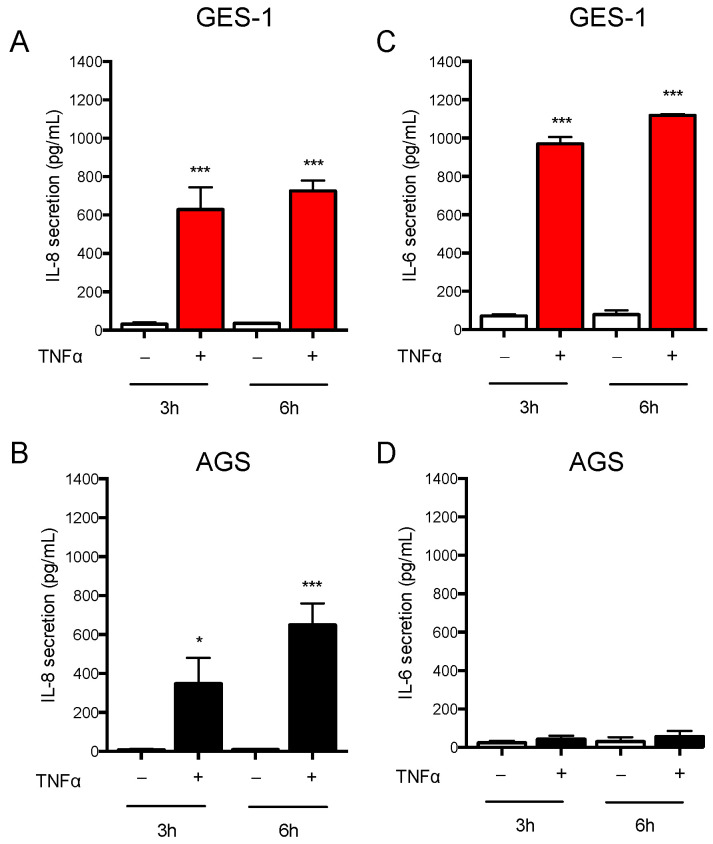
Inflammatory response to TNF in GES-1 and AGS cells. GES-1 (red bars) and AGS (black bars) cells were treated for 3 and 6 h with TNF (10 ng/mL). (**A**,**B**) IL-8 and (**C**,**D**) IL-6 secretion was measured with ELISA. Results are expressed as mean ± SEM of at least 3 experiments performed in duplicates. * *p* < 0.05, and *** *p* < 0.001 vs. untreated cells (unpaired *t*-test).

**Figure 2 ijms-24-15147-f002:**
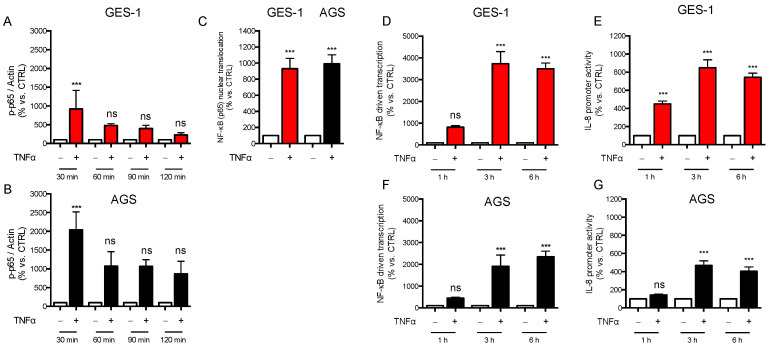
NF-κB signaling in TNF treated GES-1 and AGS cells. GES-1 (red bars) and AGS (black bars) cells were treated with TNF (10 ng/mL) for the indicated time points. (**A**,**B**) the phosphorylation state of p65 (p-p65) was assessed with western blot (representative blots are reported in Appendix A). Data are expressed as mean ± SEM of three experiments: normalized values of p-p65 bands/β-actin bands are plotted. (**C**) Nuclear translocation of p65 was evaluated with ELISA after 1 h of treatment. Data are expressed as mean ± SEM of three experiments. (**D**,**F**) NF-κB-driven transcription and (**E**,**G**) IL-8 promoter activity were evaluated following transient transfection. Results are expressed as mean ± SEM of at least three experiments performed in triplicate. n.s., not significant, *** *p* < 0.001 vs. untreated cells, to which a value of 100% was arbitrarily assigned (unpaired *t*-test).

**Figure 3 ijms-24-15147-f003:**
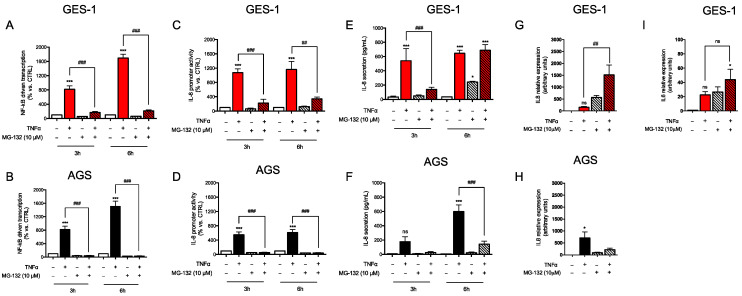
Modulation of TNF-induced inflammatory response via proteasomal inhibition. GES-1 (red bars) and AGS (black bars) cells were treated with TNF (10 ng/mL) for the indicated time points in the presence (striped bars) or absence of 10 μM MG132. (**A**,**B**) NF-κB-driven transcription, and (**C**,**D**) IL-8 promoter activity, were evaluated following transient transfection. Results are expressed as mean ± SEM of at least three experiments performed in triplicate. (**E**,**F**) IL-8 secretion was measured with ELISA. Results are expressed as mean ± SEM of at least three experiments performed in duplicates. (**G**,**H**) *CXCL8* and (**I**) *IL6* expression was evaluated with RT-qPCR. Results are expressed as mean ± SEM of at least three experiments using pooled samples obtained from three wells per each experimental condition. n.s., not significant, * *p* < 0.05, and *** *p* < 0.001 vs. untreated cells; ^##^
*p* < 0.01, ^###^
*p* < 0.001 vs. TNF (ANOVA).

**Figure 4 ijms-24-15147-f004:**
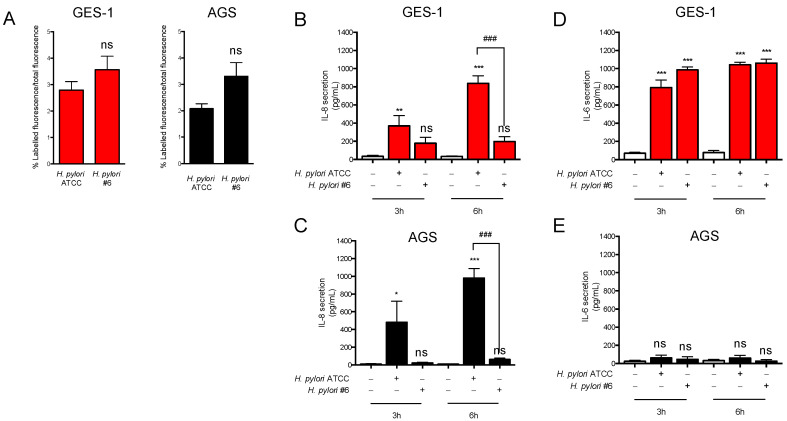
Early-stage inflammatory response upon *H. pylori* infection in GES-1 and AGS cells. GES-1 (red bars) and AGS (black bars) cells were infected with the CagA-positive (ATCC) and CagA-negative (clinical isolate #6) *H. pylori* strains (bacterium-to-cell ratio, 50:1). (**A**) Adhesion of FITC-labeled bacteria to cells was evaluated with fluorescence and expressed as percentage over total fluorescence of labeled bacteria. Results are expressed as mean ± SEM of three experiments. Total fluorescence of FITC-labeled bacteria expressed in arbitrary units is reported in the table below. (**B**,**C**) IL-8 and (**D**,**E**) IL-6 secretion were measured with ELISA. Results are expressed as mean ± SEM of 3 experiments performed in duplicates. n.s., not significant, * *p* < 0.05, ** *p* < 0.01, and *** *p* < 0.001 vs. untreated cells. ^###^
*p* < 0.001 ATCC-infected vs. #6-infected cells (ANOVA).

**Figure 5 ijms-24-15147-f005:**
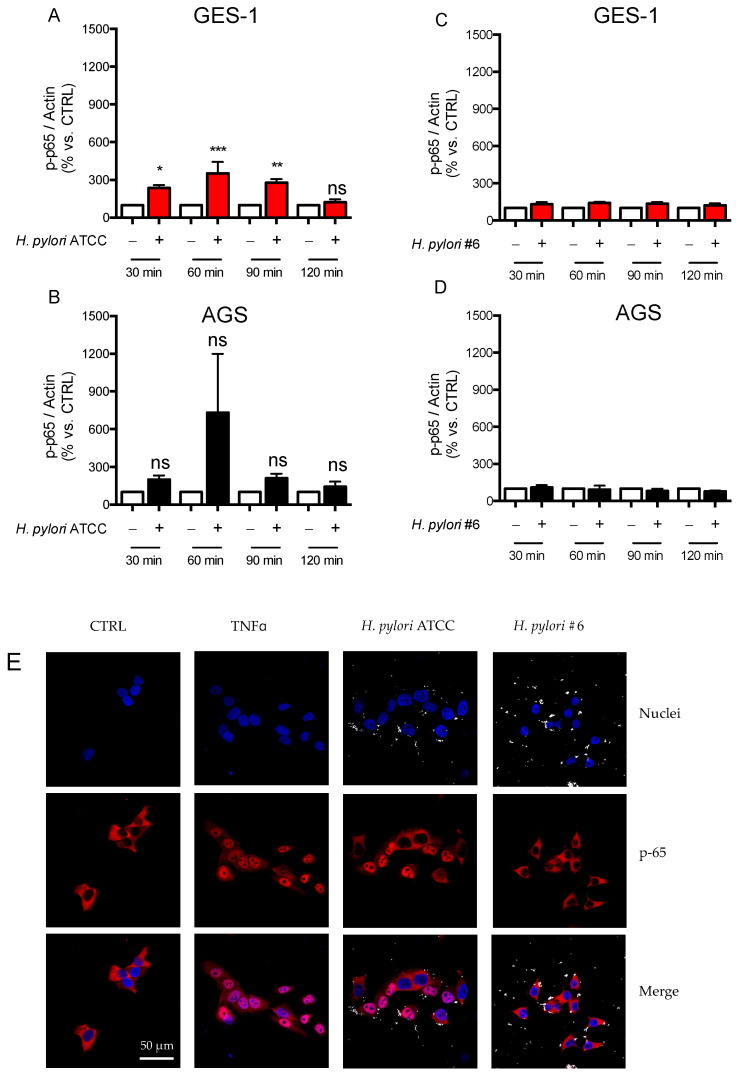
NF-κB signaling in GES-1 and AGS cells infected with *H. pylori* strains. GES-1 (red bars) and AGS (black bars) cells were infected with *H. pylori* strains (bacterium-to-cell ratio, 50:1) for the indicated time points. (**A**,**B**) the phosphorylation state of p65 (p-p65) in cells infected with the CagA-positive (ATCC) and (**C**,**D**) with the CagA-negative clinical isolate #6 (#6) strain was assessed with western blot (representative blots are reported in Appendix A). Results are expressed as mean ± SEM of at least three experiments performed in triplicate: normalized values of p-p65 bands/beta-actin bands are plotted. n.s., not significant, *** *p* < 0.001, ** *p* < 0.01, and * *p* < 0.05 (ANOVA), vs. untreated cells, to which a value of 100% was arbitrarily assigned. Intracellular localization of p65 was evaluated with confocal immunofluorescence microscopy (63X objective, bar scale 50 μm) in (**E**) GES-1 and (**F**) AGS cells treated with TNF (10 ng/mL) or infected with *H. pylori* strains. Nuclei were stained with DAPI (blue), p65 is shown in red. Brilliant dots inside and outside the cells indicate the presence of bacteria.

**Figure 6 ijms-24-15147-f006:**
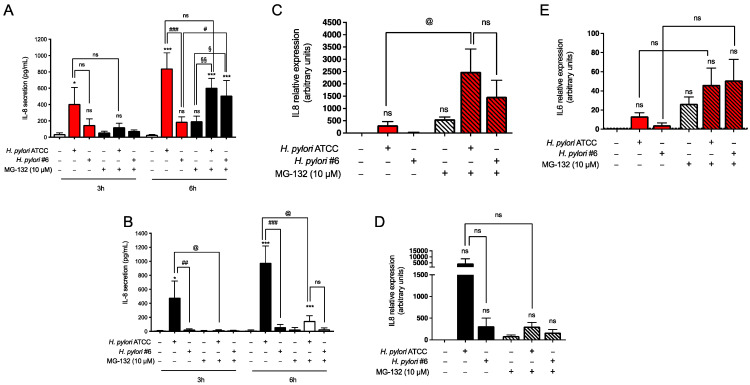
Modulation of the inflammatory response induced by *H. pylori* via proteasomal inhibition. GES-1 (red bars) and AGS (black bars) cells were infected with the CagA-positive reference strain 26,695 (ATCC) and CagA-negative clinical isolate #6 (#6) *H. pylori* strains (bacterium-to-cell ratio, 50:1) in the presence (striped bars) or absence of 10 μM MG132. (**A**,**B**) IL-8 secretion was measured with ELISA. Results are expressed as mean ± SEM of at least three experiments performed in duplicates. n.s., not significant, * *p* < 0.05, and *** *p* < 0.001 vs. untreated cells; ^#^
*p* < 0.05, ^##^
*p* < 0.01 and ^###^
*p* < 0.001 vs. #6-infected cells; ^§^
*p* < 0.05 and ^§§^
*p* < 0.01 vs. MG132 treated cells; @ *p* < 0.05 vs. ATCC-infected cells (ANOVA). (**C**,**D**) *CXCL8* and (**E**) *IL6* expression were evaluated with RT-qPCR. Results are expressed as mean ± SEM of at least three experiments using pooled samples obtained from three wells per each experimental condition. @ *p* < 0.05 vs. ATCC-infected cells (ANOVA).

**Figure 7 ijms-24-15147-f007:**
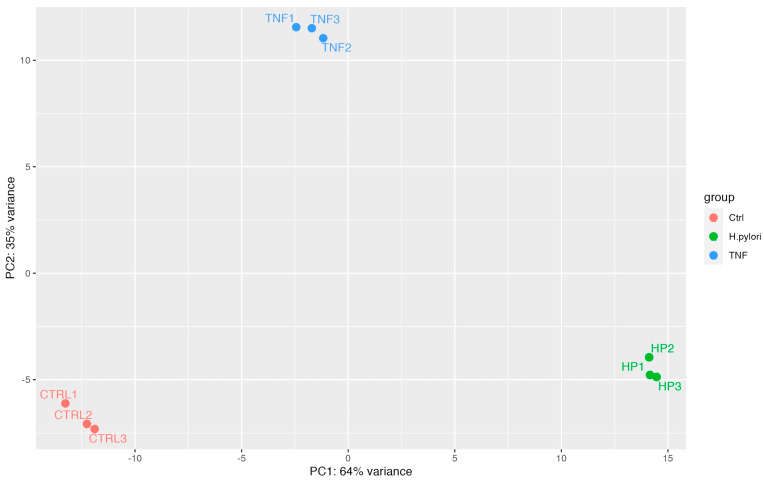
Principal component analysis of RNA-seq results. Count data obtained from sequencing were transformed by applying the log function and after normalization for the library size. Transformed data were used to perform principal component analysis (PCA) to evaluate the sample distribution and their clustering within groups representing the same condition. CTRL1-3, control cells, TNF1-3, cells treated with TNF, HP1-3, and cells infected with *H. pylori*.

**Figure 8 ijms-24-15147-f008:**
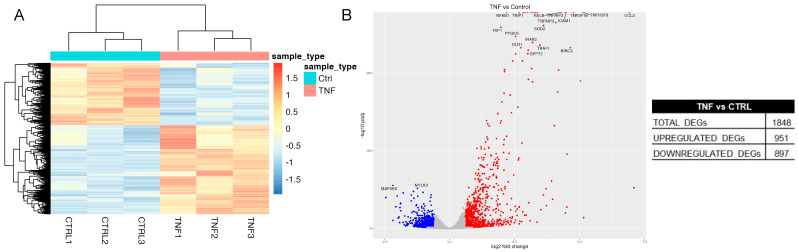
Transcriptomic analysis of TNF-treated vs. control GES-1 cells. GES-1 cells were treated with TNF (10 ng/mL) for 6 h. RNA-seq was performed as described in Materials and Methods. (**A**) Heatmap representing the distribution of differentially expressed genes (DEGs) in untreated (Ctrl) and treated (TNF) cells. (**B**) Volcano plot reporting for each gene the log2 fold change in the X-axis and the corresponding −log10 of adjusted *p* value in the Y-axis. Blue and red spots were used to indicate upregulated and downregulated genes, respectively.

**Figure 9 ijms-24-15147-f009:**
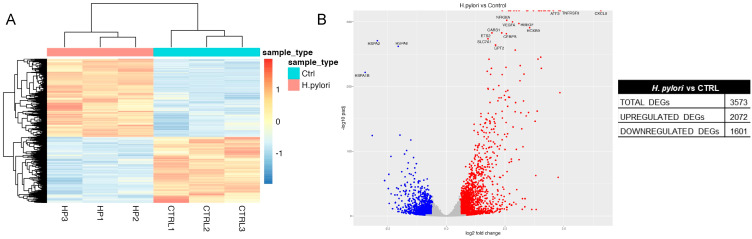
Transcriptomic analysis of *H. pylori*-infected vs. control GES-1 cells. GES-1 cells were infected with the CagA-positive (ATCC) *H. pylori* strains (bacterium-to-cell ratio, 50:1) for 6 h. RNA-seq was performed as described in Materials and Methods. (**A**) Heatmap representing the distribution of differentially expressed genes (DEGs) in control (Ctrl) and infected cells (*H. pylori*). (**B**) Volcano plot reporting for each gene the log2 fold change in the X-axis and the corresponding −log10 of adjusted *p* value in the Y-axis. Blue and red spots were used to indicate upregulated and downregulated genes, respectively.

**Figure 10 ijms-24-15147-f010:**
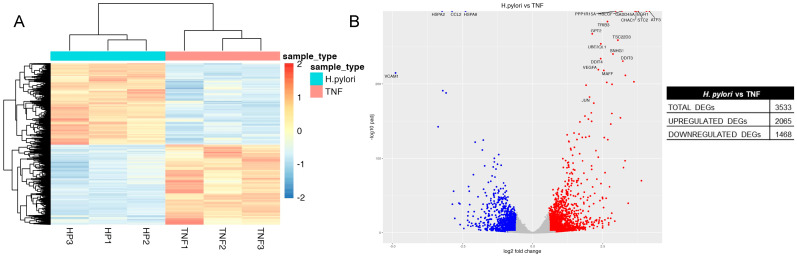
Transcriptomic analysis of *H. pylori*-infected vs. TNF-treated GES-1 cells. GES-1 cells were infected with the CagA-positive (ATCC) *H. pylori* strains (bacterium-to-cell ratio, 50:1) or treated with TNF (10 ng/mL) for 6 h. RNA-seq was performed as described in Materials and Methods. (**A**) Heatmap representing the distribution of differentially expressed genes (DEGs) in infected (*H. pylori*) and TNF-treated (TNF) cells. (**B**) Volcano plot reporting for each gene the log2 fold change in the X-axis and the corresponding −log10 of adjusted *p* value in the Y-axis. Blue and red spots were used to indicate upregulated and downregulated genes, respectively.

**Table 1 ijms-24-15147-t001:** Differentially expressed genes following 6 h-TNF treatment in GES-1 and AGS cells. (Human Inflammatory Cytokines & Receptors RT^2^ Profiler PCR Array).

	GES-1		AGS		
Gene Symbol	Nominal *p* Value	Fold Change	Nominal *p* Value	Fold Change	NF-κB Target *
*C5*	6.69 × 10^−3^	0.525	>0.05	NA	No
*CCL11*	6.88 × 10^−3^	14.655	>0.05	NA	No
*CCL2*	9.52 × 10^−5^	189.581	>0.05	NA	Yes
*CCL20*	2.95 × 10^−3^	785.064	4.78 × 10^−3^	294.747	Yes
*CCL7*	3.87 × 10^−4^	21.407	>0.05	NA	Yes
*CSF1*	1.05 × 10^−3^	10.339	3.20 × 10^−3^	3.045	No
*CSF2*	1.90 × 10^−2^	22.523	>0.05	NA	No
*CX3CL1*	1.20 × 10^−2^	10.079	>0.05	NA	No
*CXCL1*	4.50 × 10^−3^	61.963	>0.05	NA	Yes
*CXCL10*	8.55 × 10^−3^	12.729	>0.05	NA	Yes
*CXCL2 (GRO gamma)*	8.22 × 10^−4^	15.207	9.61 × 10^−5^	10.778	Yes
*CXCL3*	6.23 × 10^−4^	4.834	>0.05	NA	No
*CXCL5*	3.70 × 10^−2^	12.098	3.43 × 10^−3^	55.715	Yes
*CXCL6*	2.28 × 10^−3^	16.951	>0.05	NA	No
*CXCL8*	1.01 × 10^−3^	53.569	8.12 × 10^−4^	77.708	Yes
*IL15*	3.40 × 10^−3^	2.451	>0.05	NA	No
*IL1B*	2.99 × 10^−2^	8.938	>0.05	NA	Yes
*IL6*	1.30 × 10^−3^	28.624	>0.05	NA	Yes
*IL7*	4.56 × 10^−2^	40.039	>0.05	NA	No
*LTB*	4.79 × 10^−4^	581.378	2.56 × 10^−5^	3026.288	Yes
*MMP9*	1.47 × 10^−3^	45.932	>0.05	NA	Yes
*NAMPT*	2.52 × 10^−3^	4.103	>0.05	NA	No
*TNF*	7.11 × 10^−4^	69.712	4.56 × 10^−3^	63.705	Yes
*TNFRSF11B*	3.08 × 10^−2^	1.853	>0.05	NA	No
*VEGFA*	6.56 × 10^−3^	2.676	>0.05	NA	No

* Data extrapolated according to the Thomas Gilmore’s lab list [23] and the NF-κB target gene sets [24]. NA, not applicable, indicates that expression was not significantly changed by the treatment.

**Table 2 ijms-24-15147-t002:** Top enriched pathways with upregulated or downregulated genes in TNF-treated in comparison to untreated cells.

Term	Overlap	*p* Value	Adjusted *p* Value
**Enriched pathways with upregulated genes**			
cytokine-mediated signaling pathway (GO:0019221)	113	3.27 × 10^−38^	1.21 × 10^−34^
cellular response to cytokine stimulus (GO:0071345)	82	9.47 × 10^−26^	1.75 × 10^−22^
regulation of inflammatory response (GO:0050727)	40	3.91 × 10^−15^	4.82 x10^−12^
inflammatory response (GO:0006954)	42	7.62 × 10^−15^	6.94 × 10^−12^
cellular response to lipopolysaccharide (GO:0071222)	30	9.50 × 10^−15^	6.94 × 10^−12^
regulation of cell population proliferation (GO:0042127)	85	1.13 × 10^−14^	6.94 × 10^−12^
cellular response to tumor necrosis factor (GO:0071356)	38	1.43 × 10^−14^	7.53 × 10^−12^
positive regulation of cytokine production (GO:0001819)	49	3.33 × 10^−13^	1.42 × 10^−10^
regulation of I-kappaB kinase/NF-kappaB signaling (GO:0043122)	39	3.46 × 10^−13^	1.42 × 10^−10^
cellular response to interferon-gamma (GO:0071346)	28	5.90 × 10^−13^	2.18 × 10^−10^
**Enriched pathways with downregulated genes**			
anterior/posterior pattern specification (GO:0009952)	12	6.22 × 10^−6^	1.68 × 10^−2^
generation of neurons (GO:0048699)	22	2.05 × 10^−5^	1.92 × 10^−2^
neuron differentiation (GO:0030182	20	2.20 × 10^−5^	1.92 × 10^−2^
ameboidal-type cell migration (GO:0001667)	10	3.37 × 10^−5^	1.92 × 10^−2^
neural crest cell migration (GO:0001755)	8	8.63 × 10^−5^	3.69 × 10^−2^
negative regulation of axon extension involved in axon guidance (GO:0048843)	6	9.56 × 10^−5^	3.69 × 10^−2^
endocardial cushion development (GO:0003197)	6	1.29 × 10^−4^	4.37 × 10^−2^
regulation of branching involved in ureteric bud morphogenesis (GO:0090189)	5	2.65 × 10^−4^	6.31 × 10^−2^
axonogenesis (GO:0007409)	22	2.89 × 10^−4^	6.31 × 10^−2^
regulation of axon extension involved in axon guidance (GO:0048841)	6	2.89 × 10^−4^	6.31 × 10^−2^

**Table 3 ijms-24-15147-t003:** Top enriched pathways with upregulated or downregulated genes in *H. pylori*-infected in comparison to control cells.

Term	Overlap	*p* Value	Adjusted *p* Value
**Enriched pathways with upregulated genes**			
regulation of transcription by RNA polymerase II (GO:0006357)	337	1.35 × 10^−20^	6.17 × 10^−17^
regulation of transcription, DNA-templated (GO:0006355)	327	8.36 × 10^−17^	1.91 × 10^−13^
positive regulation of transcription by RNA polymerase II (GO:0045944)	162	7.86 × 10^−16^	1.20 × 10^−12^
positive regulation of transcription, DNA-templated (GO:0045893)	195	3.28 × 10^−15^	3.75 × 10^−12^
negative regulation of transcription by RNA polymerase II (GO:0000122)	127	6.45 × 10^−14^	5.89 × 10^−11^
negative regulation of transcription, DNA-templated (GO:0045892)	151	7.70 × 10^−11^	5.87 × 10^−8^
cytokine-mediated signaling pathway (GO:0019221)	109	1.41 × 10^−10^	9.19 × 10^−8^
cellular response to cytokine stimulus (GO:0071345)	89	5.32 × 10^−10^	3.04 × 10^−7^
regulation of gene expression (GO:0010468)	39	1.41 × 10^−9^	7.19 × 10^−7^
positive regulation of pri-miRNA transcription by RNA polymerase II (GO:1902895)	28	1.78 × 10^−9^	8.13 × 10^−7^
**Enriched pathways with downregulated genes**			
chaperone cofactor-dependent protein refolding (GO:0051085)	11	3.82 × 10^−7^	1.37 × 10^−3^
‘de novo’ posttranslational protein folding (GO:0051084)	11	3.06 × 10^−6^	5.48 × 10^−3^
response to unfolded protein (GO:0006986)	13	1.51 × 10^−5^	1.81 × 10^−2^
cellular response to unfolded protein (GO:0034620)	8	2.94 × 10^−4^	1.36 × 10^−1^
cellular response to steroid hormone stimulus (GO:0071383)	7	3.15 × 10^−4^	1.36 × 10^−1^
aminoglycan biosynthetic process (GO:0006023)	11	3.42 × 10^−4^	1.36 × 10^−1^
embryonic organ morphogenesis (GO:0048562)	10	5.68 × 10^−4^	2.00 × 10^−1^
negative regulation of viral process (GO:0048525)	13	7.28 × 10^−4^	2.18 × 10^−1^
regulation of cyclin-dependent protein kinase activity (GO:1904029)	11	8.23 × 10^−4^	2.19 × 10^−1^
odontogenesis (GO:0042476)	9	9.46 × 10^−4^	2.19 × 10^−1^

**Table 4 ijms-24-15147-t004:** Top enriched pathways with upregulated or downregulated genes in *H. pylori*-infected in comparison to TNF-treated cells.

Term	Overlap	*p* Value	Adjusted *p* Value
**Enriched pathways with upregulated genes**			
regulation of transcription, DNA-templated (GO:0006355)	316	6.04 × 10^−16^	2.55 × 10^−12^
regulation of transcription by RNA polymerase II (GO:0006357)	307	8.66 × 10^−15^	1.82 × 10^−11^
negative regulation of transcription by RNA polymerase II (GO:0000122)	118	1.13 × 10^−11^	1.58 × 10^−8^
positive regulation of transcription, DNA-templated (GO:0045893)	169	2.87 × 10^−9^	3.02 × 10^−6^
negative regulation of transcription, DNA-templated (GO:0045892)	141	4.93 × 10^−9^	4.16 × 10^−6^
positive regulation of transcription by RNA polymerase II (GO:0045944)	135	1.09 × 10^−8^	7.65 × 10^−6^
regulation of gene expression (GO:0010468)	152	4.75 × 10^−8^	2.86 × 10^−5^
intrinsic apoptotic signaling pathway in response to endoplasmic reticulum stress (GO:0070059)	12	4.10 × 10^−6^	2.16 × 10^−3^
intrinsic apoptotic signaling pathway (GO:0097193)	23	4.12 × 10^−5^	1.93 × 10^−2^
regulation of cellular macromolecule biosynthetic process (GO:2000112)	68	1.01 × 10^−4^	3.65 × 10^−2^
**Enriched pathways with downregulated genes**			
cytokine-mediated signaling pathway (GO:0019221)	80	5.59 × 10^−8^	2.25 × 10^−4^
chaperone cofactor-dependent protein refolding (GO:0051085)	11	5.05 × 10^−7^	9.87 × 10^−3^
positive regulation of I-kappaB kinase/NF-kappaB signaling (GO:0043123)	31	7.34 × 10^−7^	9.87 × 10^−3^
‘de novo’ posttranslational protein folding (GO:0051084)	11	4.00 × 10^−6^	2.70 × 10^−3^
negative regulation of viral process (GO:0048525)	17	4.22 × 10^−6^	2.70 × 10^−3^
regulation of I-kappaB kinase/NF-kappaB signaling (GO:0043122)	35	5.22 × 10^−6^	2.70 × 10^−3^
cellular response to type I interferon (GO:0071357)	16	6.68 × 10^−6^	2.70 × 10^−3^
type I interferon signaling pathway (GO:0060337)	16	6.68 × 10^−6^	2.70 × 10^−3^
mitochondrial respiratory chain complex I assembly (GO:0032981)	15	6.71 × 10^−6^	2.70 × 10^−3^
NADH dehydrogenase complex assembly (GO:0010257)	15	6.71 × 10^−6^	2.70 × 10^−3^

## Data Availability

Data are available on request from the corresponding author, S.P. (stefano.piazza@unimi.it).

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
