# Peer review of "Investigating the Molecular Mechanisms Underlying Early Response to Inflammation and *Helicobacter pylori* Infection in Human Gastric Epithelial Cells"

_ijms, 2023, doi:10.3390/ijms242015147_

Round 1

Reviewer 1 Report

Martinelli et al. investigated how non-tumoral human gastric epithelial cells (GES-1) and widely used gastric cancer AGS cell lines respond towards H. pylori during the early host response. TNF is also applied to compare with pathogen-specific responses. Through the activation of NF-κB signaling and the alteration of cytokines, several transcription factors were further identified together with transcriptome analysis. Thus, the authors identified that the immune response induced by H. Pylori affects gene transcription through transcription factors and downstream cellular pathways.

1.       In lines 94-95, please cite the references for the connection between TNF and H. pylori

2.       Fig 1, what are the statistics used here? I think should be a t-test since you’re comparing control vs. treatment.

3.       Fig 2, same concern as above, what statistics are used in here?

4.       Fig 2 A&B, do these two graphs suggest that TNFα only induces p65 phosphorylation and nuclear translocation at early treatment?

5.       In lines 156-158, have you tested the expression level of members of the Rel family in this study?

6.       In lines 195-199, cite related references here for explaining the rationale of using certain strains.

7.       Fig 4A, where is the statistic for the graph?

8.       Fig 4B, is it ANOVA here? What you indicate in the graph seems like you only compare each treatment group separately with the control.

9.       Fig 4C, same problem as above in Fig 4B.

10.   Fig 5B, you might need to explain why the 60min-treated group is non-significant by showing individual spots.

11.   For all the comparisons made between cell lines or H. pylori strains, please include information in the graph title.

12.   Fig 5 E&F, any quantification of nuclear translocation? Also scale bars are missing. 

13.   Same suggestion as above for Fig 6, include cell line name in graph title. Also why MG-132 treatment only in AGS cell line?

14.   For the transcriptomic analysis, is it possible to combine Fig 9 and Fig 10 to give the reader a more overlook view of your data?   

The overall quality of this manuscript is good, it would be great if it was further proofread by professional academic writers.   

Author Response

Dear Reviewer 1,

We thank for useful suggestions reported in the attached file. 

Reviewer 2 Report

General:

This work establishes a comparison between the immunological response of gastric lineages GES-1 and AGS against an infection stimulated by Helicobacter pylori. The authors demonstrate that GES-1 cell lines, healthy cell lines, have elevated AGS compared to carcinogenic cells.

Some points require small corrections regarding the writing of the work.

1. The microorganism Helicobacter pylori or H. pylori must be in italics. Apart from the title, throughout the text this word was not italicized.

2. I suggest that authors standardize abbreviations. Throughout the text I observed the following variations:

- MMP.9 / MMP9

- GES.1 / GES-1

- IL1B / IL-1B

- IL6 / IL-6

3. In line 55, I believe the author meant “above” instead of “aboce”.

4. Much information in the discussion is not referenced.

5. I believe this work would be of interest for the authors to explore as a discussion (https://doi.org/10.3892/ijo.2019.4775)

6. Methods 4.6, 4.7, 4.8, 4.8.1, 4.8.2 and 4.9 do not have references described.

7. In figures 2 and 5, I suggest the authors include the western blot images as supplementary. This figure is too dense.

Some points require greater attention in this work:

1. In figures 1, 3 and 6, the authors add in the caption that they work with at least 3 different biological samples. However, they report that the data was obtained based on technical duplicates. I understand that, as we are using cell cultures, the results may present many discrepancies between the variables and therefore it is recommended to perform at least one triplicate. The results present errors that are relatively high in some graphs. Because they are being demonstrated as SEM, they may be “masking” a very high standard deviation, a fact that can also be caused by duplicate analysis.

2. In figure 1, the authors demonstrate that TNF induces increased IL-8 secretion. However, in table 1 the authors do not indicate interleukin among the elevated genes. Any justification for this effect?

3. I suggest the authors add the statistical analysis used to validate the graphs to the figures.

4. The description of the results in figure 6 and its caption leave the reader confused in understanding which cell lineage the authors are exploring. Until arriving at this image there was a standard that AGS are the black bars and GES-1 are the red bars. In this image the authors mix the colors. I suggest the authors to place striped bars in all treatments using MG-132 and maintain the standard colors of the strains.

5. In figure 6B the results appear to be different from what was shown in figure 4C. In figure 4C, treatment with H. pylori ATCC induced an increase in IL-8 secretion in a proportion of ~500 pg/mL and in figure 6C the statistical difference of the same treatment was not observed and secretion decreased to ~100 pg/mL.

6. In line 311-314 the authors comment that “The fact that both strains induced IL-8 and IL-6 secretion by GES-1 cell,....” When looking at figure 6 the authors demonstrate a qPCR analysis of the IL-6 and not secretion. I believe it would be interesting to carry out a secretion analysis to confirm the results described.

7. In figure 9, I think it would be more relevant to demonstrate the analysis of GES-1 and AGS together.

Author Response

Dear Reviewer 2,

We thank for the useful suggestions attached below, point-by-point.

This work establishes a comparison between the immunological response of gastric lineages GES-1 and AGS against an infection stimulated by Helicobacter pylori. The authors demonstrate that GES-1 cell lines, healthy cell lines, have elevated AGS compared to carcinogenic cells.

Some points require small corrections regarding the writing of the work.

  1. The microorganism Helicobacter pylori or H. pylori must be in italics. Apart from the title, throughout the text this word was not italicized.

We thank the reviewer. We checked the text again for amendment.

  1. I suggest that authors standardize abbreviations. Throughout the text I observed the following variations:

- MMP.9 / MMP9

- GES.1 / GES-1

- IL1B / IL-1B

- IL6 / IL-6

We thank the reviewer for this remark and amended the incorrect abbreviations. We also verified the abbreviations for genes (i.e., IL1B and IL6) and proteins (i.e., IL-1B and IL-6) to be compliant with standard rules.

  1. In line 55, I believe the author meant “above” instead of “aboce”.

We thank the reviewer for his/her remark and amended the typo.

  1. Much information in the discussion is not referenced.

We thank the reviewer for his/her remark. We carefully checked the Discussion section and added pertinent references where required (lines 431 - 455).

  1. I believe this work would be of interest for the authors to explore as a discussion (https://doi.org/10.3892/ijo.2019.4775)

We thank the reviewer for his/her suggestion. The reference was cited in the Discussion section (line 448) (ref. 16), and we also mentioned it in the Introduction section.

  1. Methods 4.6, 4.7, 4.8, 4.8.1, 4.8.2 and 4.9 do not have references described.

As requested by the reviewer we added pertinent references where appropriate.

  1. In figures 2 and 5, I suggest the authors include the western blot images as supplementary. This figure is too dense.

We thank the reviewer for his/her suggestion. We moved the western blot images in the supplementary material (Figure S2 and Figure S3, respectively).

Some points require greater attention in this work:

  1. In figures 1, 3 and 6, the authors add in the caption that they work with at least 3 different biological samples. However, they report that the data was obtained based on technical duplicates. I understand that, as we are using cell cultures, the results may present many discrepancies between the variables and therefore it is recommended to perform at least one triplicate. The results present errors that are relatively high in some graphs. Because they are being demonstrated as SEM, they may be “masking” a very high standard deviation, a fact that can also be caused by duplicate analysis.

We took into serious account the reviewer’s comment. Data reported in figure 1 and 3 (treatment with TNF) show standard errors with values </= 25% of the measured parameter. Figure 6 reports data obtained in co-culture experiments in which the bacterial component and the virulence potential of the pathogen may contribute to biological variability. Indeed, the inflammatory response induced by the most virulent strain (ATCC) displays higher variability with respect the less virulent strain #6. We made efforts to reduce this variability by performing multiple experiments with identical experimental design. It is opinion of the authors that the final message of figure 6, i.e. the different behavior of the two cells lines and the different virulence potential of the two H. pylori strains, is not affected by variability.

  1. In figure 1, the authors demonstrate that TNF induces increased IL-8 secretion. However, in table 1 the authors do not indicate interleukin among the elevated genes. Any justification for this effect?

We thank the reviewer for his/her remark. CXCL8, the gene encoding IL-8, is indeed among the statistically up-regulated genes reported in table 1.

  1. I suggest the authors add the statistical analysis used to validate the graphs to the figures.

We thank the reviewer for his/her remarks. Figure legends were modified to include information on the statistical analysis.

  1. The description of the results in figure 6 and its caption leave the reader confused in understanding which cell lineage the authors are exploring. Until arriving at this image there was a standard that AGS are the black bars and GES-1 are the red bars. In this image the authors mix the colors. I suggest the authors to place striped bars in all treatments using MG-132 and maintain the standard colors of the strains.

We thank the reviewer for outlining the problem with the patterns of bars in this figure. We checked the colors/patterns and added labels to graphs to improve readability.

  1. In figure 6B the results appear to be different from what was shown in figure 4C. In figure 4C, treatment with H. pylori ATCC induced an increase in IL-8 secretion in a proportion of ~500 pg/mL and in figure 6C the statistical difference of the same treatment was not observed and secretion decreased to ~100 pg/mL.

We thank the reviewer for his/her remark. We amended figure 6B, time point 3 h, including the correct values.

  1. In line 311-314 the authors comment that “The fact that both strains induced IL-8 and IL-6 secretion by GES-1 cell,....” When looking at figure 6 the authors demonstrate a qPCR analysis of the IL-6 and not secretion. I believe it would be interesting to carry out a secretion analysis to confirm the results described.

We apologize for not specifying the data we referred to in the text. We added a reference to panels of figure 4 showing results on cytokine secretion by GES-1 cells infected with H. pylori strains.

  1. In figure 9, I think it would be more relevant to demonstrate the analysis of GES-1 and AGS together.

As stated at beginning of section 2.3 (lines 332), the objective of the transcriptomic analysis was “to gain further insight into the molecular pathways triggered in the early stages of H. pylori infection in GES-1 cells”. The non-tumoral origin of GES-1 cells makes them an attractive model for studying gastric inflammation with respect to cancer cell lines such as AGS.

Round 2

Reviewer 2 Report

After previous corrections, I believe that the work can be published in its current form